# Linear regression without correspondence

**Daniel Hsu**
Columbia University
New York, NY
djhsu@cs.columbia.edu

**Kevin Shi**
Columbia University
New York, NY
kshi@cs.columbia.edu

**Xiaorui Sun**
Microsoft Research
Redmond, WA
xiaoruisun@cs.columbia.edu

## Abstract

This article considers algorithmic and statistical aspects of linear regression when the correspondence between the covariates and the responses is unknown. First, a fully polynomial-time approximation scheme is given for the natural least squares optimization problem in any constant dimension. Next, in an average-case and noise-free setting where the responses exactly correspond to a linear function of i.i.d. draws from a standard multivariate normal distribution, an efficient algorithm based on lattice basis reduction is shown to exactly recover the unknown linear function in arbitrary dimension. Finally, lower bounds on the signal-to-noise ratio are established for approximate recovery of the unknown linear function by any estimator.

## 1 Introduction

Consider the problem of recovering an unknown vector $\bar{\boldsymbol{w}} \in \mathbb{R}^d$ from noisy linear measurements when the correspondence between the measurement vectors and the measurements themselves is unknown. The measurement vectors (i.e., covariates) from $\mathbb{R}^d$ are denoted by $\boldsymbol{x}_1, \boldsymbol{x}_2, \ldots, \boldsymbol{x}_n$; for each $i \in [n] := \{1, 2, \ldots, n\}$, the $i$-th measurement (i.e., response) $y_i$ is obtained using $\boldsymbol{x}_{\bar{\pi}(i)}$:

$$y_i = \bar{\boldsymbol{w}}^\top \boldsymbol{x}_{\bar{\pi}(i)} + \varepsilon_i, \quad i \in [n]. \tag{1}$$

Above, $\bar{\pi}$ is an unknown permutation on $[n]$, and the $\varepsilon_1, \varepsilon_2, \ldots, \varepsilon_n$ are unknown measurement errors.

This problem, which has been called *unlabeled sensing* [22], *linear regression with an unknown permutation* [18], and *linear regression with shuffled labels* [1], arises in many settings; see the aforementioned references for more details. In short, sensing limitations may create ambiguity in or even completely lose the ordering of measurements. The problem is also interesting because the missing correspondence makes an otherwise well-understood problem into one with very different computational and statistical properties.

**Prior works.** Unnikrishnan et al. [22] study conditions on the measurement vectors that permit recovery of any target vector $\bar{\boldsymbol{w}}$ under noiseless measurements. They show that when the entries of the $\boldsymbol{x}_i$ are drawn i.i.d. from a continuous distribution, and $n \geq 2d$, then almost surely, every vector $\bar{\boldsymbol{w}} \in \mathbb{R}^d$ is uniquely determined by noiseless correspondence-free measurements as in (1). (Under noisy measurements, it is shown that $\bar{\boldsymbol{w}}$ can be recovered when an appropriate signal-to-noise ratio tends to infinity.) It is also shown that $n \geq 2d$ is necessary for such a guarantee that holds *for all* vectors $\bar{\boldsymbol{w}} \in \mathbb{R}^d$.

Pananjady et al. [18] study statistical and computational limits on recovering the unknown permutation $\bar{\pi}$. On the statistical front, they consider necessary and sufficient conditions on the signal-to-noise ratio $\mathsf{SNR} := \|\bar{\boldsymbol{w}}\|_2^2 / \sigma^2$ when the measurement errors $(\varepsilon_i)_{i=1}^n$ are i.i.d. draws from the normal distribution $\mathrm{N}(0, \sigma^2)$ and the measurement vectors $(\boldsymbol{x}_i)_{i=1}^n$ are i.i.d. draws from the standard multivariate normal distribution $\mathrm{N}(\boldsymbol{0}, \boldsymbol{I}_d)$. Roughly speaking, exact recovery of $\bar{\pi}$ is possible via maximum likelihood

when $\mathsf{SNR} \geq n^c$ for some absolute constant $c > 0$, and approximate recovery is impossible for any method when $\mathsf{SNR} \leq n^{c'}$ for some other absolute constant $c' > 0$. On the computational front, they show that the least squares problem (which is equivalent to maximum likelihood problem)

$$\min_{\boldsymbol{w},\pi} \sum_{i=1}^{n} \left( \boldsymbol{w}^\top \boldsymbol{x}_{\pi(i)} - y_i \right)^2 \tag{2}$$

given arbitrary $\boldsymbol{x}_1, \boldsymbol{x}_2, \ldots, \boldsymbol{x}_n \in \mathbb{R}^d$ and $y_1, y_2, \ldots, y_n \in \mathbb{R}$ is NP-hard when $d = \Omega(n)$[1], but admits a polynomial-time algorithm (in fact, an $O(n \log n)$-time algorithm based on sorting) when $d = 1$.

Abid et al. [1] observe that the maximum likelihood estimator can be inconsistent for estimating $\bar{\boldsymbol{w}}$ in certain settings (including the normal setting of Pananjady et al. [18], with SNR fixed but $n \to \infty$). One of the alternative estimators they suggest is consistent under additional assumptions in dimension $d = 1$. Elhami et al. [8] give a $O(dn^{d+1})$-time algorithm that, in dimension $d = 2$, is guaranteed to approximately recover $\bar{\boldsymbol{w}}$ when the measurement vectors are chosen in a very particular way from the unit circle and the measurement errors are uniformly bounded.

**Contributions.** We make progress on both computational and statistical aspects of the problem.

1. We give an approximation algorithm for the least squares problem from (2) that, any given $(\boldsymbol{x}_i)_{i=1}^n$, $(y_i)_{i=1}^n$, and $\epsilon \in (0, 1)$, returns a solution with objective value at most $1 + \epsilon$ times that of the minimum in time $(n/\epsilon)^{O(d)}$. This a fully polynomial-time approximation scheme for any constant dimension.

2. We give an algorithm that exactly recovers $\bar{\boldsymbol{w}}$ in the measurement model from (1), under the assumption that there are no measurement errors and the covariates $(\boldsymbol{x}_i)_{i=1}^n$ are i.i.d. draws from $\mathrm{N}(\boldsymbol{0}, \boldsymbol{I}_d)$. The algorithm, which is based on a reduction to a lattice problem and employs the lattice basis reduction algorithm of Lenstra et al. [16], runs in $\mathrm{poly}(n, d)$ time when the covariate vectors $(\boldsymbol{x}_i)_{i=1}^n$ and target vector $\bar{\boldsymbol{w}}$ are appropriately quantized. This result may also be regarded as *for each*-type guarantee for exactly recovering a fixed vector $\bar{\boldsymbol{w}}$, which complements the *for all*-type results of Unnikrishnan et al. [22] concerning the number of measurement vectors needed for recovering all possible vectors.

3. We show that in the measurement model from (1) where the measurement errors are i.i.d. draws from $\mathrm{N}(0, \sigma^2)$ and the covariate vectors are i.i.d. draws from $\mathrm{N}(\boldsymbol{0}, \boldsymbol{I}_d)$, then no algorithm can approximately recover $\bar{\boldsymbol{w}}$ unless $\mathsf{SNR} \geq C \min \{1, d/\log\log(n)\}$ for some absolute constant $C > 0$. We also show that when the covariate vectors are i.i.d. draws from the uniform distribution on $[-1/2, 1/2]^d$, then approximate recovery is impossible unless $\mathsf{SNR} \geq C'$ for some other absolute constant $C' > 0$.

Our algorithms are not meant for practical deployment, but instead are intended to shed light on the computational difficulty of the least squares problem and the average-case recovery problem. Indeed, note that a naïve brute-force search over permutations requires time $\Omega(n!) = n^{\Omega(n)}$, and the only other previous algorithms (already discussed above) were restricted to $d = 1$ [18] or only had some form of approximation guarantee when $d = 2$ [8]. We are not aware of previous algorithms for the average-case problem in general dimension $d$.[2]

Our lower bounds on SNR stand in contrast to what is achievable in the classical linear regression model (where the covariate/response correspondence is known): in that model, the SNR requirement for approximately recovering $\bar{\boldsymbol{w}}$ scales as $d/n$, and hence the problem becomes easier with $n$. The lack of correspondence thus drastically changes the difficulty of the problem.

## 2 Approximation algorithm for the least squares problem

In this section, we consider the least squares problem from Equation (2). The inputs are an arbitrary matrix $\boldsymbol{X} = [\boldsymbol{x}_1 | \boldsymbol{x}_2 | \cdots | \boldsymbol{x}_n]^\top \in \mathbb{R}^{n \times d}$ and an arbitrary vector $\boldsymbol{y} = (y_1, y_2, \ldots, y_n)^\top \in \mathbb{R}^n$, and the

**Algorithm 1** Approximation algorithm for least squares problem

---

**input** Covariate matrix $\boldsymbol{X} = [\boldsymbol{x}_1|\boldsymbol{x}_2|\cdots|\boldsymbol{x}_n]^\top \in \mathbb{R}^{n\times k}$; response vector $\boldsymbol{y} = (y_1, y_2, \ldots, y_n)^\top \in \mathbb{R}^n$; approximation parameter $\epsilon \in (0,1)$.

**assume** $\boldsymbol{X}^\top \boldsymbol{X} = \boldsymbol{I}_k$.

**output** Weight vector $\hat{\boldsymbol{w}} \in \mathbb{R}^k$ and permutation matrix $\hat{\boldsymbol{\Pi}} \in \mathcal{P}_n$.

 1: Run "Row Sampling" algorithm with input matrix $\boldsymbol{X}$ to obtain a matrix $\boldsymbol{S} \in \mathbb{R}^{r\times n}$ with $r = 4k$.
 2: Let $\mathcal{B}$ be the set of vectors $\boldsymbol{b} = (b_1, b_2, \ldots, b_n)^\top \in \mathbb{R}^n$ satisfying the following: for each $i \in [n]$,
   - if the $i$-th column of $\boldsymbol{S}$ is all zeros, then $b_i = 0$;
   - otherwise, $b_i \in \{y_1, y_2, \ldots, y_n\}$.
 3: Let $c := 1 + 4(1 + \sqrt{n/(4k)})^2$.
 4: **for** each $\boldsymbol{b} \in \mathcal{B}$ **do**
 5:      Compute $\tilde{\boldsymbol{w}}_{\boldsymbol{b}} \in \arg\min_{\boldsymbol{w}\in\mathbb{R}^k} \|\boldsymbol{S}(\boldsymbol{X}\boldsymbol{w} - \boldsymbol{b})\|_2^2$, and let $r_{\boldsymbol{b}} := \min_{\boldsymbol{\Pi}\in\mathcal{P}_n} \|\boldsymbol{X}\tilde{\boldsymbol{w}}_{\boldsymbol{b}} - \boldsymbol{\Pi}^\top \boldsymbol{y}\|_2^2$.
 6:      Construct a $\sqrt{\epsilon r_{\boldsymbol{b}}/c}$-net $\mathcal{N}_{\boldsymbol{b}}$ for the Euclidean ball of radius $\sqrt{cr_{\boldsymbol{b}}}$ around $\tilde{\boldsymbol{w}}_{\boldsymbol{b}}$, so that for each $\boldsymbol{v} \in \mathbb{R}^k$ with $\|\boldsymbol{v} - \tilde{\boldsymbol{w}}_{\boldsymbol{b}}\|_2 \leq \sqrt{cr_{\boldsymbol{b}}}$, there exists $\boldsymbol{v}' \in \mathcal{N}_{\boldsymbol{b}}$ such that $\|\boldsymbol{v} - \boldsymbol{v}'\|_2 \leq \sqrt{\epsilon r_{\boldsymbol{b}}/c}$.
 7: **end for**
 8: **return** $\hat{\boldsymbol{w}} \in \arg\min_{\boldsymbol{w}\in\bigcup_{\boldsymbol{b}\in\mathcal{B}}\mathcal{N}_{\boldsymbol{b}}} \min_{\boldsymbol{\Pi}\in\mathcal{P}_n} \|\boldsymbol{X}\boldsymbol{w} - \boldsymbol{\Pi}^\top \boldsymbol{y}\|_2^2$ and $\hat{\boldsymbol{\Pi}} \in \arg\min_{\boldsymbol{\Pi}\in\mathcal{P}_n} \|\boldsymbol{X}\hat{\boldsymbol{w}} - \boldsymbol{\Pi}^\top \boldsymbol{y}\|_2^2$.

---

goal is to find a vector $\boldsymbol{w} \in \mathbb{R}^d$ and permutation matrix $\boldsymbol{\Pi} \in \mathcal{P}_n$ (where $\mathcal{P}_n$ denotes the space of $n \times n$ permutation matrices[3]) to minimize $\|\boldsymbol{X}\boldsymbol{w} - \boldsymbol{\Pi}^\top \boldsymbol{y}\|_2^2$. This problem is NP-hard in the case where $d = \Omega(n)$ [18] (see also Appendix A). We give an approximation scheme that, for any $\epsilon \in (0,1)$, returns a $(1+\epsilon)$-approximation in time $(n/\epsilon)^{O(k)} + \text{poly}(n,d)$, where $k := \text{rank}(\boldsymbol{X}) \leq \min\{n,d\}$.

We assume without loss of generality that $\boldsymbol{X} \in \mathbb{R}^{n\times k}$ and $\boldsymbol{X}^\top \boldsymbol{X} = \boldsymbol{I}_k$. This is because we can always replace $\boldsymbol{X}$ with its matrix of left singular vectors $\boldsymbol{U} \in \mathbb{R}^{n\times k}$, obtained via singular value decomposition $\boldsymbol{X} = \boldsymbol{U}\boldsymbol{\Sigma}\boldsymbol{V}^\top$, where $\boldsymbol{U}^\top \boldsymbol{U} = \boldsymbol{V}^\top \boldsymbol{V} = \boldsymbol{I}_k$ and $\boldsymbol{\Sigma} \succ 0$ is diagonal. A solution $(\boldsymbol{w}, \boldsymbol{\Pi})$ for $(\boldsymbol{U}, \boldsymbol{y})$ has the same cost as the solution $(\boldsymbol{V}\boldsymbol{\Sigma}^{-1}\boldsymbol{w}, \boldsymbol{\Pi})$ for $(\boldsymbol{X}, \boldsymbol{y})$, and a solution $(\boldsymbol{w}, \boldsymbol{\Pi})$ for $(\boldsymbol{X}, \boldsymbol{y})$ has the same cost as the solution $(\boldsymbol{\Sigma}\boldsymbol{V}^\top \boldsymbol{w}, \boldsymbol{\Pi})$ for $(\boldsymbol{U}, \boldsymbol{y})$.

## 2.1 Algorithm

Our approximation algorithm, shown as Algorithm 1, uses a careful enumeration to beat the naïve brute-force running time of $\Omega(|\mathcal{P}_n|) = \Omega(n!)$. It uses as a subroutine a "Row Sampling" algorithm of Boutsidis et al. [5] (described in Appendix B), which has the following property.

**Theorem 1** (Specialization of Theorem 12 in [5])**.** *There is an algorithm ("Row Sampling") that, given any matrix $\boldsymbol{A} \in \mathbb{R}^{n\times k}$ with $n \geq k$, returns in $\text{poly}(n,k)$ time a matrix $\boldsymbol{S} \in \mathbb{R}^{r\times n}$ with $r = 4k$ such that the following hold.*

 1. *Every row of $\boldsymbol{S}$ has at most one non-zero entry.*

 2. *For every $\boldsymbol{b} \in \mathbb{R}^n$, every $\boldsymbol{w}' \in \arg\min_{\boldsymbol{w}\in\mathbb{R}^k} \|\boldsymbol{S}(\boldsymbol{A}\boldsymbol{w} - \boldsymbol{b})\|_2^2$ satisfies $\|\boldsymbol{A}\boldsymbol{w}' - \boldsymbol{b}\|_2^2 \leq c \cdot \min_{\boldsymbol{w}\in\mathbb{R}^k} \|\boldsymbol{A}\boldsymbol{w} - \boldsymbol{b}\|_2^2$ for $c = 1 + 4(1 + \sqrt{n/(4k)})^2 = O(n/k)$.*

The matrix $\boldsymbol{S}$ returned by Row Sampling determines a (weighted) subset of $O(k)$ rows of $\boldsymbol{A}$ such that solving a (ordinary) least squares problem (with any right-hand side $\boldsymbol{b}$) on this subset of rows and corresponding right-hand side entries yields a $O(n/k)$-approximation to the least squares problem over all rows and right-hand side entries. Row Sampling does not directly apply to our problem because (1) it does not minimize over permutations of the right-hand side, and (2) the approximation factor is too large. However, we are able to use it to narrow the search space in our problem.

An alternative to Row Sampling is to simply enumerate all subsets of $k$ rows of $\boldsymbol{X}$. This is justified by a recent result of Dereziński and Warmuth [7], which shows that for any right-hand side $\boldsymbol{b} \in \mathbb{R}^n$, using "volume sampling" [3] to choose a matrix $\boldsymbol{S} \in \{0,1\}^{k\times k}$ (where each row has one non-zero entry) gives a similar guarantee as that of Row Sampling, except with the $O(n/k)$ factor replaced by $k+1$ in expectation.

## 2.2 Analysis

The approximation guarantee of Algorithm 1 is given in the following theorem.

**Theorem 2.** *Algorithm 1 returns $\hat{\boldsymbol{w}} \in \mathbb{R}^k$ and $\hat{\boldsymbol{\Pi}} \in \mathcal{P}_n$ satisfying*

$$\left\| \boldsymbol{X}\hat{\boldsymbol{w}} - \hat{\boldsymbol{\Pi}}^\top \boldsymbol{y} \right\|_2^2 \;\leq\; (1+\epsilon) \min_{\boldsymbol{w} \in \mathbb{R}^k, \boldsymbol{\Pi} \in \mathcal{P}_n} \left\| \boldsymbol{X}\boldsymbol{w} - \boldsymbol{\Pi}^\top \boldsymbol{y} \right\|_2^2 \; .$$

*Proof.* Let $\mathrm{opt} := \min_{\boldsymbol{w},\boldsymbol{\Pi}} \| \boldsymbol{X}\boldsymbol{w} - \boldsymbol{\Pi}^\top \boldsymbol{y} \|_2^2$ be the optimal cost, and let $(\boldsymbol{w}_\star, \boldsymbol{\Pi}_\star)$ denote a solution achieving this cost. The optimality implies that $\boldsymbol{w}_\star$ satisfies the normal equations $\boldsymbol{X}^\top \boldsymbol{X}\boldsymbol{w}_\star = \boldsymbol{X}^\top \boldsymbol{\Pi}_\star^\top \boldsymbol{y}$. Observe that there exists a vector $\boldsymbol{b}_\star \in \mathcal{B}$ satisfying $\boldsymbol{S}\boldsymbol{b}_\star = \boldsymbol{S}\boldsymbol{\Pi}_\star^\top \boldsymbol{y}$. By Theorem 1 and the normal equations, the vector $\tilde{\boldsymbol{w}}_{\boldsymbol{b}_\star}$ and cost value $r_{\boldsymbol{b}_\star}$ satisfy

$$\mathrm{opt} \;\leq\; r_{\boldsymbol{b}_\star} \;\leq\; \left\| \boldsymbol{X}\tilde{\boldsymbol{w}}_{\boldsymbol{b}_\star} - \boldsymbol{\Pi}_\star^\top \boldsymbol{y} \right\|_2^2 \;=\; \left\| \boldsymbol{X}(\tilde{\boldsymbol{w}}_{\boldsymbol{b}_\star} - \boldsymbol{w}_\star) \right\|_2^2 + \mathrm{opt} \;\leq\; c \cdot \mathrm{opt} \; .$$

Moreover, since $\boldsymbol{X}^\top \boldsymbol{X} = \boldsymbol{I}_k$, we have that $\| \tilde{\boldsymbol{w}}_{\boldsymbol{b}_\star} - \boldsymbol{w}_\star \|_2 \leq \sqrt{(c-1)\,\mathrm{opt}} \leq \sqrt{cr_{\boldsymbol{b}_\star}}$. By construction of $\mathcal{N}_{\boldsymbol{b}_\star}$, there exists $\boldsymbol{w} \in \mathcal{N}_{\boldsymbol{b}_\star}$ satisfying $\| \boldsymbol{w} - \boldsymbol{w}_\star \|_2^2 = \| \boldsymbol{X}(\boldsymbol{w} - \boldsymbol{w}_\star) \|_2^2 \leq \epsilon r_{\boldsymbol{b}_\star}/c \leq \epsilon\,\mathrm{opt}$. For this $\boldsymbol{w}$, the normal equations imply

$$\min_{\boldsymbol{\Pi} \in \mathcal{P}_n} \| \boldsymbol{X}\boldsymbol{w} - \boldsymbol{\Pi}^\top \boldsymbol{y} \|_2^2 \;\leq\; \| \boldsymbol{X}\boldsymbol{w} - \boldsymbol{\Pi}_\star^\top \boldsymbol{y} \|_2^2 \;=\; \| \boldsymbol{X}(\boldsymbol{w} - \boldsymbol{w}_\star) \|_2^2 + \mathrm{opt} \;\leq\; (1+\epsilon)\,\mathrm{opt} \; .$$

Therefore, the solution returned by Algorithm 1 has cost no more than $(1+\epsilon)\,\mathrm{opt}$. $\qquad\square$

By the results of Pananjady et al. [18] for maximum likelihood estimation, our algorithm enjoys recovery guarantees for $\bar{\boldsymbol{w}}$ and $\bar{\pi}$ when the data come from the Gaussian measurement model (1). However, the approximation guarantee also holds for worst-case inputs without generative assumptions.

**Running time.** We now consider the running time of Algorithm 1. There is the initial cost for singular value decomposition (as discussed at the beginning of the section), and also for "Row Sampling"; both of these take $\mathrm{poly}(n,d)$ time. For the rest of the algorithm, we need to consider the size of $\mathcal{B}$ and the size of the net $\mathcal{N}_{\boldsymbol{b}}$ for each $\boldsymbol{b} \in \mathcal{B}$. First, we have $|\mathcal{B}| \leq n^r = n^{O(k)}$, since $\boldsymbol{S}$ has only $4k$ rows and each row has at most a single non-zero entry. Next, for each $\boldsymbol{b} \in \mathcal{B}$, we construct the $\delta$-net $\mathcal{N}_{\boldsymbol{b}}$ (for $\delta := \sqrt{\epsilon r_{\boldsymbol{b}}/c}$) by constructing a $\delta/\sqrt{k}$-net for the $\ell_\infty$-ball of radius $\sqrt{cr_{\boldsymbol{b}}}$ centered at $\tilde{\boldsymbol{w}}_{\boldsymbol{b}}$ (using an appropriate axis-aligned grid). This has size $|\mathcal{N}_{\boldsymbol{b}}| \leq (4c^2 k/\epsilon)^{k/2} = (n/\epsilon)^{O(k)}$. Finally, each $\arg\min_{\boldsymbol{w} \in \mathbb{R}^k}$ computation takes $O(nk^2)$ time, and each $(\arg)\min_{\boldsymbol{\Pi} \in \mathcal{P}_n}$ takes $O(nk + n\log n)$ time [18] (also see Appendix B). So, the overall running time is $(n/\epsilon)^{O(k)} + \mathrm{poly}(n,d)$.

## 3  Exact recovery algorithm in noiseless Gaussian setting

To counter the intractability of the least squares problem in (2) confronted in Section 2, it is natural to explore distributional assumptions that may lead to faster algorithms. In this section, we consider the noiseless measurement model where the $(\boldsymbol{x}_i)_{i=1}^n$ are i.i.d. draws from $\mathrm{N}(\boldsymbol{0}, \boldsymbol{I}_d)$ (as in [18]). We give an algorithm that exactly recovers $\bar{\boldsymbol{w}}$ with high probability when $n \geq d + 1$. The algorithm runs in $\mathrm{poly}(n,d)$-time when $(\boldsymbol{x}_i)_{i=1}^n$ and $\bar{\boldsymbol{w}}$ are appropriately quantized.

It will be notationally simpler to consider $n+1$ covariate vectors and responses

$$y_i \;=\; \bar{\boldsymbol{w}}^\top \boldsymbol{x}_{\bar{\pi}(i)}, \quad i = 0, 1, \ldots, n. \tag{3}$$

Here, $(\boldsymbol{x}_i)_{i=0}^n$ are $n+1$ i.i.d. draws from $\mathrm{N}(\boldsymbol{0}, \boldsymbol{I}_d)$, the unknown permutation $\bar{\pi}$ is over $\{0, 1, \ldots, n\}$, and the requirement of at least $d+1$ measurements is expressed as $n \geq d$.

In fact, we shall consider a variant of the problem in which we are given one of the values of the unknown permutation $\bar{\pi}$. Without loss of generality, assume we are given that $\bar{\pi}(0) = 0$. Solving this variant of the problem suffices because there are only $n+1$ possible values of $\bar{\pi}(0)$: we can try them all, incurring just a factor $n+1$ in the computation time. So henceforth, we just consider $\bar{\pi}$ as an unknown permutation on $[n]$.

---

**Algorithm 2** Find permutation

---

**input** Covariate vectors $\boldsymbol{x}_0, \boldsymbol{x}_1, \boldsymbol{x}_2, \ldots, \boldsymbol{x}_n$ in $\mathbb{R}^d$; response values $y_0, y_1, y_2, \ldots, y_n$ in $\mathbb{R}$; confidence parameter $\delta \in (0, 1)$; lattice parameter $\beta > 0$.

**assume** there exists $\bar{\boldsymbol{w}} \in \mathbb{R}^d$ and permutation $\bar{\pi}$ on $[n]$ such that $y_i = \bar{\boldsymbol{w}}^\top \boldsymbol{x}_{\bar{\pi}(i)}$ for each $i \in [n]$, and that $y_0 = \bar{\boldsymbol{w}}^\top \boldsymbol{x}_0$.

**output** Permutation $\hat{\pi}$ on $[n]$ or failure.

1: Let $\boldsymbol{X} = [\boldsymbol{x}_1 | \boldsymbol{x}_2 | \cdots | \boldsymbol{x}_n]^\top \in \mathbb{R}^{n \times d}$, and its pseudoinverse be $\boldsymbol{X}^\dagger = [\tilde{\boldsymbol{x}}_1 | \tilde{\boldsymbol{x}}_2 | \cdots | \tilde{\boldsymbol{x}}_n]$.
2: Create Subset Sum instance with $n^2$ source numbers $c_{i,j} := y_i \tilde{\boldsymbol{x}}_j^\top \boldsymbol{x}_0$ for $(i, j) \in [n] \times [n]$ and target sum $y_0$.
3: Run Algorithm 3 with Subset Sum instance and lattice parameter $\beta$.
4: **if** Algorithm 3 returns a solution $\mathcal{S} \subseteq [n] \times [n]$ **then**
5:     **return** any permutation $\hat{\pi}$ on $[n]$ such that $\hat{\pi}(i) = j$ implies $(i, j) \in \mathcal{S}$.
6: **else**
7:     **return** failure.
8: **end if**

---

**Algorithm 3** Lagarias and Odlyzko [12] subset sum algorithm

---

**input** Source numbers $\{c_i\}_{i \in \mathcal{I}} \subset \mathbb{R}$; target sum $t \in \mathbb{R}$; lattice parameter $\beta > 0$.

**output** Subset $\hat{\mathcal{S}} \subseteq \mathcal{I}$ or failure.

1: Construct lattice basis $\boldsymbol{B} \in \mathbb{R}^{(|\mathcal{I}|+2) \times (|\mathcal{I}|+1)}$ where

$$\boldsymbol{B} := \left[ \begin{array}{c} \boldsymbol{I}_{|\mathcal{I}|+1} \\ \hline \beta t \mid -\beta c_i : i \in \mathcal{I} \end{array} \right] \ \in \ \mathbb{R}^{(|\mathcal{I}|+2) \times (|\mathcal{I}|+1)} .$$

2: Run basis reduction [e.g., 16] to find non-zero lattice vector $\boldsymbol{v}$ of length at most $2^{|\mathcal{I}|/2} \cdot \lambda_1(\boldsymbol{B})$.
3: **if** $\boldsymbol{v} = z(1, \boldsymbol{\chi}_{\hat{\mathcal{S}}}^\top, 0)^\top$, with $z \in \mathbb{Z}$ and $\boldsymbol{\chi}_{\hat{\mathcal{S}}} \in \{0, 1\}^{\mathcal{I}}$ is characteristic vector for some $\hat{\mathcal{S}} \subseteq \mathcal{I}$ **then**
4:     **return** $\hat{\mathcal{S}}$.
5: **else**
6:     **return** failure.
7: **end if**

---

## 3.1 Algorithm

Our algorithm, shown as Algorithm 2, is based on a reduction to the Subset Sum problem. An instance of Subset Sum is specified by an unordered collection of source numbers $\{c_i\}_{i \in \mathcal{I}} \subset \mathbb{R}$, and a target sum $t \in \mathbb{R}$. The goal is to find a subset $\mathcal{S} \subseteq \mathcal{I}$ such that $\sum_{i \in \mathcal{S}} c_i = t$. Although Subset Sum is NP-hard in the worst case, it is tractable for certain structured instances [12, 9]. We prove that Algorithm 2 constructs such an instance with high probability. A similar algorithm based on such a reduction was recently used by Andoni et al. [2] for a different but related problem.

Algorithm 2 proceeds by (i) solving a Subset Sum instance based on the covariate vectors and response values (using Algorithm 3), and (ii) constructing a permutation $\hat{\pi}$ on $[n]$ based on the solution to the Subset Sum instance. With the permutation $\hat{\pi}$ in hand, we (try to) find a solution $\boldsymbol{w} \in \mathbb{R}^d$ to the system of linear equations $y_i = \boldsymbol{w}^\top \boldsymbol{x}_{\hat{\pi}(i)}$ for $i \in [n]$. If $\hat{\pi} = \bar{\pi}$, then there is a unique such solution almost surely.

## 3.2 Analysis

The following theorem is the main recovery guarantee for Algorithm 2.

**Theorem 3.** *Pick any $\delta \in (0, 1)$. Suppose $(\boldsymbol{x}_i)_{i=0}^n$ are i.i.d. draws from $\mathrm{N}(\boldsymbol{0}, \boldsymbol{I}_d)$, and $(y_0)_{i=1}^n$ follow the noiseless measurement model from (3) for some $\bar{\boldsymbol{w}} \in \mathbb{R}^d$ and permutation $\bar{\pi}$ on $[n]$ (and $\bar{\pi}(0) = 0$), and that $n \geq d$. Furthermore, suppose Algorithm 2 is run with inputs $(\boldsymbol{x}_i)_{i=0}^n$, $(y_i)_{i=0}^n$, $\delta$, and $\beta$, and also that $\beta \geq 2^{n^2}/\varepsilon$ where $\varepsilon$ is defined in Equation (8). With probability at least $1 - \delta$, Algorithm 2 returns $\hat{\pi} = \bar{\pi}$.*

**Remark 1.** The value of $\varepsilon$ from Equation (8) is directly proportional to $\|\bar{\boldsymbol{w}}\|_2$, and Algorithm 2 requires a lower bound on $\varepsilon$ (in the setting of the lattice parameter $\beta$). Hence, it suffices to determine

a lower bound on $\|\bar{\boldsymbol{w}}\|_2$. Such a bound can be obtained from the measurement values: a standard tail bound (Lemma 6 in Appendix C) shows that with high probability, $\sqrt{\sum_{i=1}^n y_i^2/(2n)}$ is a lower bound on $\|\bar{\boldsymbol{w}}\|_2$, and is within a constant factor of it as well.

**Remark 2.** Algorithm 2 strongly exploits the assumption of noiseless measurements, which is expected given the SNR lower bounds of Pananjady et al. [18] for recovering $\bar{\pi}$. The algorithm, however, is also very brittle and very likely fails in the presence of noise.

**Remark 3.** The recovery result does not contradict the results of Unnikrishnan et al. [22], which show that a collection of $2d$ measurement vectors are necessary for recovering all $\bar{\boldsymbol{w}}$, even in the noiseless measurement model of (3). Indeed, our result shows that for a *fixed* $\bar{\boldsymbol{w}} \in \mathbb{R}^d$, with high probability $d + 1$ measurements in the model of (3) suffice to permit exactly recovery of $\bar{\boldsymbol{w}}$, but this same set of measurement vectors (when $d + 1 < 2d$) will fail for some other $\bar{\boldsymbol{w}}'$.

The proof of Theorem 3 is based on the following theorem—essentially due to Lagarias and Odlyzko [12] and Frieze [9]—concerning certain structured instances of Subset Sum that can be solved using the lattice basis reduction algorithm of Lenstra et al. [16]. Given a basis $\boldsymbol{B} = [\boldsymbol{b}_1|\boldsymbol{b}_2|\cdots|\boldsymbol{b}_k] \in \mathbb{R}^{m \times k}$ for a lattice

$$\mathcal{L}(\boldsymbol{B}) := \left\{ \sum_{i=1}^k z_i \boldsymbol{b}_i : z_1, z_2, \ldots, z_k \in \mathbb{Z} \right\} \subset \mathbb{R}^m,$$

this algorithm can be used to find a non-zero vector $\boldsymbol{v} \in \mathcal{L}(\boldsymbol{B}) \setminus \{\boldsymbol{0}\}$ whose length is at most $2^{(k-1)/2}$ times that of the shortest non-zero vector in the lattice

$$\lambda_1(\boldsymbol{B}) := \min_{\boldsymbol{v} \in \mathcal{L}(\boldsymbol{B}) \setminus \{\boldsymbol{0}\}} \|\boldsymbol{v}\|_2 .$$

**Theorem 4** ([12, 9]). *Suppose the Subset Sum instance specified by source numbers $\{c_i\}_{i \in \mathcal{I}} \subset \mathbb{R}$ and target sum $t \in \mathbb{R}$ satisfy the following properties.*

1. *There is a subset $\mathcal{S}^\star \subseteq \mathcal{I}$ such that $\sum_{i \in \mathcal{S}^\star} c_i = t$.*

2. *Define $R := 2^{|\mathcal{I}|/2}\sqrt{|\mathcal{S}^\star| + 1}$ and $\mathcal{Z}_R := \{(z_0, \boldsymbol{z}) \in \mathbb{Z} \times \mathbb{Z}^{\mathcal{I}} : 0 < z_0^2 + \sum_{i \in \mathcal{I}} z_i^2 \leq R^2\}$. There exists $\varepsilon > 0$ such that $|z_0 \cdot t - \sum_{i \in \mathcal{I}} z_i \cdot c_i| \geq \varepsilon$ for each $(z_0, \boldsymbol{z}) \in \mathcal{Z}_R$ that is not an integer multiple of $(1, \boldsymbol{\chi}^\star)$, where $\boldsymbol{\chi}^\star \in \{0, 1\}^{\mathcal{I}}$ is the characteristic vector for $\mathcal{S}^\star$.*

*Let $\boldsymbol{B}$ be the lattice basis $\boldsymbol{B}$ constructed by Algorithm 3, and assume $\beta \geq 2^{|\mathcal{I}|/2}/\varepsilon$. Then every non-zero vector in the lattice $\Lambda(\boldsymbol{B})$ with length at most $2^{|\mathcal{I}|/2}$ times the length of the shortest non-zero vector in $\Lambda(\boldsymbol{B})$ is an integer multiple of the vector $(1, \boldsymbol{\chi}_{\mathcal{S}^\star}, 0)$, and the basis reduction algorithm of Lenstra et al. [16] returns such a non-zero vector.*

The Subset Sum instance constructed in Algorithm 2 has $n^2$ source numbers $\{c_{i,j} : (i, j) \in [n] \times [n]\}$ and target sum $y_0$. We need to show that it satisfies the two conditions of Theorem 4.

Let $\mathcal{S}_{\bar{\pi}} := \{(i, j) : \bar{\pi}(i) = j\} \subset [n] \times [n]$, and let $\bar{\boldsymbol{\Pi}} = (\bar{\Pi}_{i,j})_{(i,j) \in [n] \times [n]} \in \mathcal{P}_n$ be the permutation matrix with $\bar{\Pi}_{i,j} := \mathbb{1}\{\bar{\pi}(i) = j\}$ for all $(i, j) \in [n] \times [n]$. Note that $\bar{\boldsymbol{\Pi}}$ is the "characteristic vector" for $\mathcal{S}_{\bar{\pi}}$. Define $R := 2^{n^2/2}\sqrt{n + 1}$ and

$$\mathcal{Z}_R := \left\{ (z_0, \boldsymbol{Z}) \in \mathbb{Z} \times \mathbb{Z}^{n \times n} : 0 < z_0^2 + \sum_{1 \leq i,j \leq n} Z_{i,j}^2 \leq R^2 \right\} .$$

A crude bound shows that $|\mathcal{Z}_R| \leq 2^{O(n^4)}$.

The following lemma establishes the first required property in Theorem 4.

**Lemma 1.** *The random matrix $\boldsymbol{X}$ has rank $d$ almost surely, and the subset $\mathcal{S}_{\bar{\pi}}$ satisfies $y_0 = \sum_{(i,j) \in \mathcal{S}_{\bar{\pi}}} c_{i,j}$.*

*Proof.* That $\boldsymbol{X}$ has rank $d$ almost surely follows from the fact that the probability density of $\boldsymbol{X}$ is supported on all of $\mathbb{R}^{n \times d}$. This implies that $\boldsymbol{X}^\dagger \boldsymbol{X} = \sum_{j=1}^n \tilde{\boldsymbol{x}}_j \boldsymbol{x}_j^\top = \boldsymbol{I}_d$, and

$$y_0 = \sum_{j=1}^n \boldsymbol{x}_0^\top \tilde{\boldsymbol{x}}_j \boldsymbol{x}_j^\top \bar{\boldsymbol{w}} = \sum_{1 \leq i,j \leq n} \boldsymbol{x}_0^\top \tilde{\boldsymbol{x}}_j \cdot y_i \cdot \mathbb{1}\{\bar{\pi}(i) = j\} = \sum_{1 \leq i,j \leq n} c_{i,j} \cdot \mathbb{1}\{\bar{\pi}(i) = j\}. \quad \square$$

The next lemma establishes the second required property in Theorem 4. Here, we use the fact that the Frobenius norm $\left\|z_0\bar{\mathbf{\Pi}} - \mathbf{Z}\right\|_F$ is at least one whenever $(z_0, \mathbf{Z}) \in \mathbb{Z} \times \mathbb{Z}^{n \times n}$ is not an integer multiple of $(1, \bar{\mathbf{\Pi}})$.

**Lemma 2.** *Pick any $\eta, \eta' > 0$ such that $3|\mathcal{Z}_R|\eta + \eta' < 1$. With probability at least $1 - 3|\mathcal{Z}_R|\eta - \eta'$, every $(z_0, \mathbf{Z}) \in \mathcal{Z}_R$ with $\mathbf{Z} = (Z_{i,j})_{(i,j) \in [n] \times [n]}$ satisfies*

$$\left| z_0 \cdot y_0 - \sum_{i,j} Z_{i,j} \cdot c_{i,j} \right| \geq \frac{(\pi/4) \cdot \sqrt{(d-1)/n} \cdot \eta^{2 + \frac{1}{d-1}}}{\left( \sqrt{n} + \sqrt{d} + \sqrt{2\ln(1/\eta')} \right)^2} \cdot \left\| z_0\bar{\mathbf{\Pi}} - \mathbf{Z} \right\|_F \cdot \|\bar{\boldsymbol{w}}\|_2 \,.$$

*Proof.* By Lemma 1, the matrix $\bar{\mathbf{\Pi}}$ satisfies $y_0 = \sum_{i,j} \bar{\Pi}_{i,j} \cdot c_{i,j}$. Fix any $(z_0, \mathbf{Z}) \in \mathcal{Z}_R$ with $\mathbf{Z} = (Z_{i,j})_{(i,j) \in [n] \times [n]}$. Then

$$z_0 \cdot y_0 - \sum_{i,j} Z_{i,j} \cdot c_{i,j} = \sum_{i,j} (z_0 \cdot \bar{\Pi}_{i,j} - Z_{i,j}) \cdot \boldsymbol{x}_0^\top \tilde{\boldsymbol{x}}_j \cdot \bar{\boldsymbol{w}}^\top \boldsymbol{x}_{\bar{\pi}(i)} \,.$$

Using matrix and vector notations, this can be written compactly as the inner product $\boldsymbol{x}_0^\top (\boldsymbol{X}^\dagger (z_0\bar{\mathbf{\Pi}} - \boldsymbol{Z})^\top \bar{\mathbf{\Pi}} \boldsymbol{X} \bar{\boldsymbol{w}})$. Since $\boldsymbol{x}_0 \sim \mathrm{N}(\mathbf{0}, \boldsymbol{I}_d)$ and is independent of $\boldsymbol{X}$, the distribution of the inner product is normal with mean zero and standard deviation equal to $\|\boldsymbol{X}^\dagger (z_0\bar{\mathbf{\Pi}} - \boldsymbol{Z})^\top \bar{\mathbf{\Pi}} \boldsymbol{X} \bar{\boldsymbol{w}}\|_2$. By Lemma 7 (in Appendix C), with probability at least $1 - \eta$,

$$\left| \boldsymbol{x}_0^\top \left( \boldsymbol{X}^\dagger (z_0\bar{\mathbf{\Pi}} - \boldsymbol{Z})^\top \bar{\mathbf{\Pi}} \boldsymbol{X} \bar{\boldsymbol{w}} \right) \right| \geq \|\boldsymbol{X}^\dagger (z_0\bar{\mathbf{\Pi}} - \boldsymbol{Z})^\top \bar{\mathbf{\Pi}} \boldsymbol{X} \bar{\boldsymbol{w}}\|_2 \cdot \sqrt{\frac{\pi}{2}} \cdot \eta \,. \tag{4}$$

Observe that $\boldsymbol{X}^\dagger = (\boldsymbol{X}^\top \boldsymbol{X})^{-1} \boldsymbol{X}^\top$ since $\boldsymbol{X}$ has rank $d$ by Lemma 1, so

$$\|\boldsymbol{X}^\dagger (z_0\bar{\mathbf{\Pi}} - \boldsymbol{Z})^\top \bar{\mathbf{\Pi}} \boldsymbol{X} \bar{\boldsymbol{w}}\|_2 \geq \frac{\|\boldsymbol{X}^\top (z_0\bar{\mathbf{\Pi}} - \boldsymbol{Z})^\top \bar{\mathbf{\Pi}} \boldsymbol{X} \bar{\boldsymbol{w}}\|_2}{\|\boldsymbol{X}\|_2^2} \,. \tag{5}$$

By Lemma 4 (in Appendix C), with probability at least $1 - \eta'$,

$$\|\boldsymbol{X}\|_2^2 \leq \left( \sqrt{n} + \sqrt{d} + \sqrt{2\ln(1/\eta')} \right)^2 \,. \tag{6}$$

And by Lemma 9 (in Appendix C), with probability at least $1 - 2\eta$,

$$\|\boldsymbol{X}^\top (z_0\bar{\mathbf{\Pi}} - \boldsymbol{Z})^\top \bar{\mathbf{\Pi}} \boldsymbol{X} \bar{\boldsymbol{w}}\|_2 \geq \left\| (z_0\bar{\mathbf{\Pi}} - \boldsymbol{Z})^\top \bar{\mathbf{\Pi}} \right\|_F \cdot \|\bar{\boldsymbol{w}}\|_2 \cdot \sqrt{\frac{(d-1)\pi}{8n}} \cdot \eta^{1+1/(d-1)} \,. \tag{7}$$

Since $\bar{\mathbf{\Pi}}$ is orthogonal, we have that $\|(z_0\bar{\mathbf{\Pi}} - \boldsymbol{Z})^\top \bar{\mathbf{\Pi}}\|_F = \|z_0\bar{\mathbf{\Pi}} - \boldsymbol{Z}\|_F$. Combining this with (4), (5), (6), and (7), and union bounds over all $(z_0, \boldsymbol{Z}) \in \mathcal{Z}_R$ proves the claim. $\square$

*Proof of Theorem 3.* Lemma 1 and Lemma 2 (with $\eta' := \delta/2$ and $\eta := \delta/(6|\mathcal{Z}_R|)$) together imply that with probability at least $1 - \delta$, the source numbers $\{c_{i,j} : (i,j) \in [n] \times [n]\}$ and target sum $y_0$ satisfy the conditions of Theorem 4 with

$$\mathcal{S}^\star := \{(i,j) \in [n] \times [n] : \bar{\pi}(i) = j\} \,,$$

$$\varepsilon := \frac{(\pi/4) \cdot \sqrt{(d-1)/n} \cdot (\delta/(6|\mathcal{Z}_R|))^{2 + \frac{1}{d-1}}}{\left( \sqrt{n} + \sqrt{d} + \sqrt{2\ln(2/\delta)} \right)^2} \cdot \|\bar{\boldsymbol{w}}\|_2 \geq 2^{-\operatorname{poly}(n, \log(1/\delta))} \cdot \|\bar{\boldsymbol{w}}\|_2 \,. \tag{8}$$

Thus, in this event, Algorithm 3 (with $\beta$ satisfying $\beta \geq 2^{n^2/2}/\varepsilon$) returns $\hat{\mathcal{S}} = \mathcal{S}^\star$, which uniquely determines the permutation $\hat{\pi} = \bar{\pi}$ returned by Algorithm 2. $\square$

**Running time.** The basis reduction algorithm of Lenstra et al. [16] is iterative, with each iteration primarily consisting of Gram-Schmidt orthogonalization and another efficient linear algebraic process called "size reduction". The total number of iterations required is

$$O\left( \frac{k(k+1)}{2} \log\left( \sqrt{k} \cdot \frac{\max_{i \in [k]} \|\boldsymbol{b}_i\|_2}{\lambda_1(\boldsymbol{B})} \right) \right) \,.$$

In our case, $k = n^2$ and $\lambda_1(\boldsymbol{B}) = \sqrt{n+1}$; and by Lemma 10 (in Appendix C), each of the basis vectors constructed has squared length at most $1 + \beta^2 \cdot \text{poly}(d, \log(n), 1/\delta) \cdot \|\bar{\boldsymbol{w}}\|_2^2$. Using the tight setting of $\beta$ required in Theorem 3, this gives a $\text{poly}(n, d, \log(1/\delta))$ bound on the total number of iterations as well as on the total running time.

However, the basis reduction algorithm requires both arithmetic and rounding operations, which are typically only available for finite precision rational inputs. Therefore, a formal running time analysis would require the idealized real-valued covariate vectors $(\boldsymbol{x}_i)_{i=0}^n$ and unknown target vector $\bar{\boldsymbol{w}}$ to be quantized to finite precision values. This is doable, and is similar to using a discretized Gaussian distribution for the distribution of the covariate vectors (and assuming $\bar{\boldsymbol{w}}$ is a vector of finite precision values), but leads to a messier analysis incomparable to the setup of previous works. Nevertheless, it would be desirable to find a different algorithm that avoids lattice basis reduction that still works with just $d + 1$ measurements.

## 4 Lower bounds on signal-to-noise for approximate recovery

In this section, we consider the measurement model from (1) where $(\boldsymbol{x}_i)_{i=1}^n$ are i.i.d. draws from either $N(\mathbf{0}, \boldsymbol{I}_d)$ or the uniform distribution on $[-1/2, 1/2]^d$, and $(\varepsilon_i)_{i=1}^n$ are i.i.d. draws from $N(0, \sigma^2)$. We establish lower bounds on the signal-to-noise ratio (SNR),

$$\mathsf{SNR} \; = \; \frac{\|\bar{\boldsymbol{w}}\|_2^2}{\sigma^2} \,,$$

required by any estimator $\hat{\boldsymbol{w}} = \hat{\boldsymbol{w}}((\boldsymbol{x}_i)_{i=1}^n, (y_i)_{i=1}^n)$ for $\bar{\boldsymbol{w}}$ to approximately recover $\bar{\boldsymbol{w}}$ in expectation. The estimators may have *a priori* knowledge of the values of $\|\bar{\boldsymbol{w}}\|_2$ and $\sigma^2$.

**Theorem 5.** *Assume $(\varepsilon_i)_{i=1}^n$ are i.i.d. draws from $N(0, \sigma^2)$.*

1. *There is an absolute constant $C > 0$ such that the following holds. If $n \geq 3$, $d \geq 22$, $(\boldsymbol{x}_i)_{i=1}^n$ are i.i.d. draws from $N(\mathbf{0}, \boldsymbol{I}_d)$, $(y_i)_{i=1}^n$ follow the measurement model from (1), and*

$$\mathsf{SNR} \; \leq \; C \cdot \min\left\{ \frac{d}{\log\log(n)}, \, 1 \right\},$$

   *then for any estimator $\hat{\boldsymbol{w}}$, there exists some $\bar{\boldsymbol{w}} \in \mathbb{R}^d$ such that*

$$\mathbb{E}\left[\|\hat{\boldsymbol{w}} - \bar{\boldsymbol{w}}\|_2\right] \; \geq \; \frac{1}{24}\|\bar{\boldsymbol{w}}\|_2 \,.$$

2. *If $(\boldsymbol{x}_i)_{i=1}^n$ are i.i.d. draws from the uniform distribution on $[-1/2, 1/2]^d$, and $(y_i)_{i=1}^n$ follow the measurement model from (1), and*

$$\mathsf{SNR} \; \leq \; 2 \,,$$

   *then for any estimator $\hat{\boldsymbol{w}}$, there exists some $\bar{\boldsymbol{w}} \in \mathbb{R}^d$ such that*

$$\mathbb{E}\left[\|\hat{\boldsymbol{w}} - \bar{\boldsymbol{w}}\|_2\right] \; \geq \; \frac{1}{2}\left(1 - \frac{1}{\sqrt{2}}\right)\|\bar{\boldsymbol{w}}\|_2 \,.$$

Note that in the classical linear regression model where $y_i = \bar{\boldsymbol{w}}^\top \boldsymbol{x}_i + \varepsilon_i$ for $i \in [n]$, the maximum likelihood estimator $\hat{\boldsymbol{w}}_{\mathsf{mle}}$ satisfies $\mathbb{E}\|\hat{\boldsymbol{w}}_{\mathsf{mle}} - \bar{\boldsymbol{w}}\|_2 \leq C\sigma\sqrt{d/n}$, where $C > 0$ is an absolute constant. Therefore, the SNR requirement to approximately recover $\bar{\boldsymbol{w}}$ up to (say) Euclidean distance $\|\bar{\boldsymbol{w}}\|_2 / 24$ is $\mathsf{SNR} \geq 24^2 C d/n$. Compared to this setting, Theorem 5 implies that with the measurement model of (1), the SNR requirement (as a function of $n$) is at substantially higher ($d/\log\log(n)$ in the normal covariate case, or a constant not even decreasing with $n$ in the uniform covariate case).

For the normal covariate case, Pananjady et al. [18] show that if $n > d$, $\epsilon < \sqrt{n}$, and

$$\mathsf{SNR} \; \geq \; n^{c \cdot \frac{n}{n-d} + \epsilon} \,,$$

then the maximum likelihood estimator $(\hat{\boldsymbol{w}}_{\mathsf{mle}}, \hat{\pi}_{\mathsf{mle}})$ (i.e., any minimizer of (2)) satisfies $\hat{\pi}_{\mathsf{mle}} = \bar{\pi}$ with probability at least $1 - c'n^{-2\epsilon}$. (Here, $c > 0$ and $c' > 0$ are absolute constants.) It is straightforward to see that, on the same event, we have $\|\hat{\boldsymbol{w}}_{\mathsf{mle}} - \bar{\boldsymbol{w}}\|_2 \leq C\sigma\sqrt{d/n}$ for some absolute

constant $C > 0$. Therefore, the necessary and sufficient conditions on SNR for approximate recovery of $\bar{w}$ lie between $C'd/\log\log(n)$ and $n^{C''}$ (for absolute constants $C', C'' > 0$). Narrowing this range remains an interesting open problem.

A sketch of the proof in the normal covariate case is as follows. Without loss of generality, we restrict attention to the case where $\bar{w}$ is a unit vector. We construct a $1/\sqrt{2}$-packing of the unit sphere in $\mathbb{R}^d$; the target $\bar{w}$ will be chosen from from this set. Observe that for any distinct $u, u' \in U$, each of $(x_i^\top u)_{i=1}^n$ and $(x_i^\top u')_{i=1}^n$ is an i.i.d. sample from $N(0,1)$ of size $n$; we prove that they therefore determine empirical distributions that are close to each other in Wasserstein-2 distance with high probability. We then prove that conditional on this event, the resulting distributions of $(y_i)_{i=1}^n$ under $\bar{x} = u$ and $\bar{x} = u'$ (for any pair $u, u' \in U$) are close in Kullback-Leibler divergence. Hence, by (a generalization of) Fano's inequality [see, e.g., 11], no estimator can determine the correct $u \in U$ with high probability.

The proof for the uniform case is similar, using $U = \{e_1, -e_1\}$ where $e_1 = (1, 0, \ldots, 0)^\top$. The full proof of Theorem 5 is given in Appendix D.

## Acknowledgments

We are grateful to Ashwin Pananjady, Michał Dereziński, and Manfred Warmuth for helpful discussions. DH was supported in part by NSF awards DMR-1534910 and IIS-1563785, a Bloomberg Data Science Research Grant, and a Sloan Research Fellowship. XS was supported in part by a grant from the Simons Foundation (#320173 to Xiaorui Sun). This work was done in part while DH and KS were research visitors and XS was a research fellow at the Simons Institute for the Theory of Computing.

## Footnotes

[1]Pananjady et al. [18] prove that PARTITION reduces to the problem of deciding if the optimal value of (2) is zero or non-zero. Note that PARTITION is weakly, but not strongly, NP-hard: it admits a pseudo-polynomial-time algorithm [10, Section 4.2]. In Appendix A, we prove that the least squares problem is strongly NP-hard by reduction from 3-PARTITION (which is strongly NP-complete [10, Section 4.2.2]).

[2]A recent algorithm of Pananjady et al. [19] exploits a similar average-case setting but only for a somewhat easier variant of the problem where more information about the unknown correspondence is provided.

[3]Each permutation matrix $\boldsymbol{\Pi} \in \mathcal{P}_n$ corresponds to a permutation $\pi$ on $[n]$; the $(i,j)$-th entry of $\boldsymbol{\Pi}$ is one if $\pi(i) = j$ and is zero otherwise.

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
