[Reviews · NeurIPS 2017]

Reviewer 1



The topic of linear regression with unknown correspondence is interesting and should be studied intensively. The main difficulty is the high complexity of the set of permutations when the number n of measurement vectors from R^d is large, and the design problem is to overcome this difficulty by efficient algorithms. When the response is noiseless with \sigma =0, the exact recovery problem is pretty similar to that in compressive sensing, and nice statistical analysis is carried out for the algorithm reduced to the Subset Sum problem. The approximate recovery problem with \sigma >0 is much more involved, and only lower dimensional cases with d=1, 2 were considered in the literature. Here the authors present a polynomial time method by means of a Row Sampling algorithm and provide some approximation analysis. Lower bounds for the approximate recovery are also provided to illustrate that the signal-to-noise ratio needs to be large for efficient recoveries. This part is not fully developed and the used generalized Fano method is rather standard. In general, the topic is interesting and the obtained results are excellent.

Reviewer 2



The article "Linear regression without correspondence" considers the problem of estimation in linear regression model in specific situation where the correspondence between the covariates and the responses is unknown. The authors propose the fully polynomial algorithms for the solution of least squares problem and also study the statistical lower bounds. The main emphasis of the article is on the construction of fully polynomial algorithms for least squares problem in noisy and noiseless case, while previously only the algorithms with exponential complexity were known for the cases with dimension d > 1. For the noisy case the authors propose the algorithm which gives a solution of least squares problem with any prespecified accuracy. For noiseless case another algorithm is proposed, which gives the exact solution of the least squares problem. Finally, the authors prove the upper bound for the range of signal to noise ratio values for which the consistent parameter recovery is impossible. In general, the proposed algorithms, though being not practical, help to make an important step in understanding of computational limits in the linear regression without correspondence. The statistical analysis is limited to lower bound while for the upper bound the authors refer to the paper by Pananjady et al. (2016). What puzzles me a lot is that provided lower bound for parameters recovery is d / log log n, while in Pananjady et al. (2016) the lower bound for permutation recovery is proved to be n^c for some c > 0. Moreover, in another paper Pananjady et al. (2017) the constant upper bound on prediction risk is proved. While all these results consider the different quantities to be analyzed, it is seems that the fundamental statistical limits in this problem is far from being well understood. To sum up, I think that the paper presents a strong effort in the interesting research direction and the results are sound. However, I believe that the main impact of the paper is on computational side, while some additional statistical analysis is highly desired for this problem. Also I believe that such type of paper, which includes involved theoretical analysis as well as algorithms which are more technical than intuitive, is much more suitable for full journal publication than for short conference paper.

Reviewer 3



summary: In this paper, the authors studied theoretical properties of linear regression problems without correspondence between the covariate and response variables. Since the least squares problem including permutation operations is computationally intractable, an approximation algorithm was proposed. The degree of the approximation was theoretically clarified. Furthermore, under the noiseless setting, the authors proposed a learning algorithm that achieves the exact recovery with high probability. Also, a lower bound of the estimation accuracy for noisy data was presented. review: In this paper, the linear regression problems without correspondence are intensively studied. I checked all proofs except the supplementary appendix and I believe that proofs in the main part of the paper are mathematically sound. The main results are heavily depends on some previous works such as Theorems 1 and 4, and the algorithm 2 is fragile to the noise as the authors pointed out. However, I think that the contribution of this paper is still significant. Especially, the lower bound shown in Section 4 is interesting and it will be worthwhile for the theorist in NIPS community. This paper focus only on theoretical aspects of the present problem and totally lacks applications and numerical experiments. This may be a small flaw of this paper. other comments: - l.90: I believe that w in "satisfies \|Aw-b\|" should be w'. - Running time analysis was provided in l.117 and l.220 and it showed that the computation cost is of the order poly(n,d) or n^O(k) or so. This is still high and computationally depending in practice. Probably, there may be a gap between theoretical runtime and the practical computation cost. Some comments on it would be nice.